# A Low-Phase-Noise 8 GHz Linear-Band Sub-Millimeter-Wave Phase-Locked Loop in 22 nm FD-SOI CMOS

**DOI:** 10.3390/mi14051010

**Published:** 2023-05-08

**Authors:** Mamady Kebe, Mihai Sanduleanu

**Affiliations:** 1School of Electrical Engineering and Computer Science (EECS), University of Ottawa, Ottawa, ON K1N 6N5, Canada; 2System on Chip Center, Khalifa University of Science and Technology, Abu Dhabi P.O. Box 127788, United Arab Emirates; mihai.sanduleanu@ku.ac.ae

**Keywords:** phase-locked loop, voltage-controlled oscillator, frequency divider, phase noise, sub-millimeter-wave

## Abstract

Low-phase noise and wideband phased-locked loops (PLLs) are crucial for high-data rate communication and imaging systems. Sub-millimeter-wave (sub-mm-wave) PLLs typically exhibit poor performance in terms of noise and bandwidth due to higher device parasitic capacitances, among other reasons. In this regard, a low-phase-noise, wideband, integer-N, type-II phase-locked loop was implemented in the 22 nm FD-SOI CMOS process. The proposed wideband linear differential tuning I/Q voltage-controlled oscillator (VCO) achieves an overall frequency range of 157.5–167.5 GHz with 8 GHz linear tuning and a phase noise of −113 dBc/Hz @ 100 KHz. Moreover, the fabricated PLL produces a phase noise less than −103 dBc/Hz @ 1 KHz and −128 dBc/Hz @ 100 KHz, corresponding to the lowest phase noise generated by a sub-millimeter-wave PLL to date. The measured RF output saturated power and DC power consumption of the PLL are 2 dBm and 120.75 mW, respectively, whereas the fabricated chip comprising a power amplifier and an integrated antenna occupies an area of 1.25 × 0.9 mm^2^.

## 1. Introduction

Many recent satellite communication, radar, and imaging systems use W and G bands to transmit and receive signals at high data rates [1,2,3]. To increase the data rates and sensitivity of these systems, wideband low-phase-noise and precise millimeter-wave (mm-wave) and sub-millimeter-wave (sub-mm-wave) signal sources are required [4]. Despite the use of III–V or SiGe technologies for applications at these upper millimeter-wave (mm-wave) frequency bands, many recent system implementations were performed with the CMOS technology to minimize their overall cost.

However, the main challenge posed in the design of communication systems based on CMOS technology at these frequencies is the poor performance of the essential building blocks of the phase-locked loops (PLLs), such as voltage-controlled oscillators (VCOs) with low phase noise and wide tuning ranges. All performance measures of the VCOs, including the amplitude of the output signal, tuning range, phase noise, and power consumption, unavoidably deteriorate as the operating frequency approaches the transit frequency (fT), which is limited by the parasitic capacitance of the transistors. In addition, the performance is further degraded by the high interconnection loss and low-quality factor (Q) of varactors. Many mm-wave and sub-mm-wave VCOs are based on frequency multiplications [4,5,6,7]. Although the frequency multiplication technique may slightly improve the phase noise [7], it exhibits a narrower locking range, as well as higher power and area consumption. Moreover, high-frequency PLLs use frequency synthesizers (FSs) with a low-frequency reference signal. Since voltage-controlled oscillators (VCOs) operate at high frequency, a phase comparison with a crystal reference oscillator carried out by a phase/frequency detector (PFD) should also be performed at high frequency. However, there is a speed limitation of the PFD, which is based on digital blocks at mm-wave and sub-mm-wave frequencies. In this regard, it is necessary to use frequency division to bring down the VCO output signal frequency to the desired frequency suitable for the PFD operation. Nevertheless, conventional frequency division is performed by a current-mode logic (CML), which is limited in operating frequency and presents high non-linearity [8]. For these reasons, several recent mm-wave FSs use injection-locked frequency dividers (ILFDs) or Miller frequency dividers [1,9]. Nevertheless, the implemented PLLs have narrow locking ranges, limiting their bandwidths and operating frequencies. Moreover, a CMOS-compatible spintronic-based signal generator was proposed in [10]. Although the proposed device size is in the nanoscale with two degrees of freedom tuning ability, it requires high-gain amplification to bring the output power closer to 0 dBm, therefore increasing the overall power consumption and non-linearities. Furthermore, an optical solution was proposed in [11] to design a low-phase-noise wideband-modulated waveform generator centered at 40 GHz for synthetic-aperture radars (SARs) applications. Besides its CMOS incompatibility, the designed oscillator presents a large footprint unlike integrated RF PLLs.

In this work, a type II PLL with wide locking range operating in the sub-mm-wave frequency is implemented in the 22 nm FD SOI CMOS process. The frequency performance of the PLL is improved by the design of wideband quadrature VCO along with a wide locking range ILFD with PMOS input injection.

The rest of this paper is organized as follows: Section 2 discusses the system architecture and the circuit implementation of the essential building blocks; Section 3 shows the measurement results; and Section 4 summarizes the work.

## 2. System Architecture and Circuit Design

The proposed type II sub-mm-wave PLL is given in Figure 1 and is composed of a balanced I/Q VCO, injection-locked divider (÷4) and ÷64 CML frequency dividers, a phase/frequency detector (PFD), charge pump (CP), and a loop filter (LF). An external signal generator is used to provide a reference signal between 4.984 GHz to 5.234 GHz. This signal is then divided by eight using a ÷2 CML divider chain to provide the PFD reference signal between 623.047 MHz and 654.297 MHz. The linear-tuning output frequency of the PLL ranges from 159.5 GHz to 167.5 GHz, corresponding to the VCO output frequency range.

The output signal of the I/Q VCO is amplified by a three-stage CS power amplifier with a power gain of 10 dB, a maximum saturated power of 10 dBm, and OP1 dB of 7 dBm.

### 2.1. I/Q VCO

The circuit configuration of the wideband I/Q VCO is shown in Figure 2. The resonator, which determines the output frequency of the VCO, was designed using grounded coplanar waveguides TL1–TL2, TL7–TL8, and the parasitic capacitances of transistors M1–M8. Transistors M1–M4 form the core of the VCO, which consists of two cross-coupled oscillators. Quadrature signal generation is achieved by the connection of transistors M5–M8. Consequently, balanced in-phase (I) and quadrature (Q) signals are obtained at the output of the VCO. The VCO outputs are isolated from the subsequent stages using buffers built by transistors M9–M12 along with grounded coplanar waveguides (CPWGs) TL5–TL8. A DC current of I_0_ = 5 mA is used to bias the VCO through a current mirror configuration. The size of the biasing transistors is small enough to keep the resonance frequency of the resonator unchanged and maximize the output voltage swing. As a result, an output differential peak voltage of 550 mV is obtained for a supply voltage VDD of 1.5 VDC. Due to the significance of the parasitic capacitances of MOSFETs at mm-wave frequencies, the current tuning is more practical than varactor tuning in the mm-wave VCO design. The MOSFET gate–source capacitance is related to its drain–source voltage or drain current [12]. Therefore, a change in the bias current results in a change in the parasitic capacitances of M1–M12, which dominate the total capacitance of the resonator. Thus, a current-tuning network is constructed using the transistors M13–M16 and resistors R1–R2. A DC voltage is applied to the gates of the tuning transistors, producing drain currents that are added to the fixed bias current from the current mirror to feed the QVCO core transistors. The usage of the tuning transistors is crucial to isolate the VCO core from the tuning DC voltage source as well as to ensure the oscillation by reducing power loss trough tuning. Moreover, a differential tuning voltage is applied across VCTRL+ and VCTRL− to obtain higher linearity and a wider tuning frequency range. To obtain the positive tuning, for instance, VCTRL− is maintained at a fixed low DC voltage point and VCTRL+ is tuned between VCTRL− and VDD. Negative tuning is obtained by swapping the VCTRL+ and VCTRL− from the positive tuning scenario. As a result, a wide frequency range ∆ω = 10 GHz and a phase noise less than −100 dBc/Hz @ 100KHz were obtained at the output of the I/Q VCO.

### 2.2. PLL Divider Chain

The I/Q VCO output frequency, being in the order of 160 GHz, is too large for the PFD, which is made of digital blocks. Therefore, a frequency division is necessary to bring the I/Q VCO output frequency down to the center frequency of 638.67 MHz, suitable for phase and frequency comparison. The divider chain is composed of two different divider types, namely the injection-locked frequency divider (ILFD) and the current-mode logic (CML) divider. First, a network of 2 ÷ 2 ILFD (÷4) was designed to convert the center frequency f_0_ from 163.5 GHz to 40.875 GHz, followed by the CML divider network. The ÷2 ILFD is depicted in Figure 3a and is based on injecting a current at f_0_ in the middle point of the cross-coupled pair of M1 and M2. CPWG lines TL3 and TL4 are used in the resonator circuit along with the parasitic capacitances of M1, M2, and M5 instead of inductors for the first two dividers from the VCO output. This allows a wider locking range compared to the inductor-based divider.

The sinusoidal voltage signal from the I/Q VCO buffer is fed to the gate of the transistor M5, which produces a tail current 
$i_{i}=I_{inj}cos\left(\omega_{0}t+\varphi \right)+I_{0}$
, where 
$I_{inj}<I_{0}$
. The condition 
$I_{inj}<I_{0}$
 is necessary to ensure that a net positive current is injected. Denoting *Q* as the quality factor of the LC tank, and 
$\omega_{0}=2\times \frac{1}{\sqrt{LC}}$
 and 
$I_{osc}$
 as its resonating frequency and current amplitude of oscillation, respectively, the synchronization range of the ILFD can be approximated as follows [13]:
(1)
$${∆\omega }_{0}\approx \frac{\omega_{0}}{2Q}.\frac{4}{\pi }.\frac{I_{inj}}{I_{osc}}$$


As the current is injected through the PMOS transistor M5, 
$I_{inj}=g_{m5}V_{inj}$
, where the injected voltage 
$V_{inj}$
 corresponds to the output voltage of the VCO and 
$g_{m5}$
 is the transconductance of M5. In consequence, the synchronization range is proportional to the injected voltage amplitude 
$V_{inj}$
, which is the peak voltage of the preceding VCO stage. Meanwhile, the PMOS injection offers easier voltage transfer (matching) and current conversion from the VCO, as well as lower process capacitance, resulting in a wider locking range of the divider resonator. Additionally, a differential-to-single-ended buffer is employed to isolate the VCO from the first ÷2 divider.

Next, a six-stage ÷2 CML divider (÷64) succeeds the ILFD network to bring the center frequency from 40.875 GHz down to 638.67 MHz. The conventional CML divider utilizes two D-latches with resistive elements in a master–slave configuration [5]. Here, a simplified input-unbalanced D-latch with no resistive load is used (as seen in Figure 3b). The divider uses two latches cross-coupling to allow clock division. The first latch is composed of transistors M1–M4, whereas the second is formed by M5–M8. The latch transistors M3–M4 and M7–M8 also provide resistive loading. The divider input is given through the current source I_0_, which is implemented by an NMOS transistor with W/L twice that of M1–M2 or M5–M6. The designed improved CML divider exhibits high speed and full voltage swing with large bandwidth.

### 2.3. Other PLL Blocks

Other blocks include the phase/frequency detector (PFD), the charge pump (CP), and the loop filter (LF). The output of the frequency divider chain is applied at the DIV and 
$\overset{-}{DIV}$
 inputs and the 5 GHz reference at REF and 
$\overset{-}{REF}$
 inputs of the PFD. As represented in Figure 4a, the three-state PFD is composed of two D-latches and an AND gate. When the reference signal is “high” while the divider signal is “low”, the outputs UP-
$\overset{-}{UP}$
 and DOWN-
$\overset{-}{DOWN}$
 of the D-flip-flops (DFF) are forced to be “high” and “low”, respectively. This causes the VCO to increase its output frequency, matching the reference frequency after subsequent frequency divisions. Similarly, a “low” state of the reference signal will induce a decrease in the VCO/divider output in order to match the reference frequency. The “zero” state is created when the reference and divider signals are equal in phase and frequency, leading to a reset of the DFFs through the AND gate. Meanwhile, when the two input signals of the PFD converge, a fast reset path causes the DFFs to enter state “zero” before any charge is transmitted to the succeeding CP stage. This is called the “dead-zone” and creates spurs in the PLL output. Therefore, the minimum pulse width of the PFD is limited to the reset path delay [14]. The dead-zone may be reduced by using inverter-generated delay blocks following the AND gate [15] or by employing a NOR gate in place of the AND gate [16]. In this work, an AND gate is formed by using a symmetric NAND gate followed by an inverter, allowing the reduction of the dead-zone and resulting in a wideband PFD. The PFD outputs are applied to a differential charge pump (shown in Figure 4b). The transistors M1–M4 are driven differentially by UP-UPn and DOWN-DOWNn signals from the PFD. Therefore, the two I_0_ current sources from GND and VDD will be steered in transistors M1–M4 and the differential loop filter (LF) based on R1, C1 and R2, C2. The resistors R2 connected to VCM will set the common-mode voltage at the outputs of the CP to VCM. The differential architecture has several advantages, including an insensitivity to the PMOS and NMOS switch mismatches and improved speed and voltage range [17].

Moreover, a three-stage common-source power amplifier (PA) followed by an on-chip bowtie antenna was designed to amplify the PLL output in a transmitter-antenna configuration (as seen in Figure 1). The schematic of the PA is depicted in Figure 4c. The first and second stages are composed of the duplicate transistor M1 as well as the coplanar waveguides TL2 and TL3, respectively providing voltage amplification. The last stage is formed by M2 and TL3 and performs power amplification. interstage capacitors C3 and C4 are used to suppress the DC signal from one stage to the other. The size of M2 is twice that of M1 to handle the output current of 10 mA. A common bias current of 5 mA is used to bias the class A amplifier with a DC supply voltage VDD of 1.5 V through a feeding network formed by the transistor Mb and the resistor R1. As a result, a total DC power consumption of 30 mW is generated. The high-pass network consisting of capacitors C1 and C2 and the coplanar waveguide TL1 was designed to match the input of the PA to 50 Ω. The gain of the PA ranges from 6.5 dB to 10 dB within the 159.5–167.5 GHz band. Meanwhile, the wideband bowtie antenna has a bandwidth of 32 GHz ranging from 140 GHz to 172 GHz, with a peak gain and reflection coefficient of 7 dBi and −25 dB at 160 GHz. The designed PLL is applied in a millimeter-wave transmitter to facilitate the performance measurement procedure.

## 3. Experimental Results

The chip (shown in Figure 5) was realized in the 22 nm CMOS FDSOI process and has an area size of 1.25 mm × 0.9 mm. The integrated bowtie antenna has inter-layers of metal fills for metal density DRC rules. This does not affect the 7 dB gain of the antenna and its efficiency.

To measure the PLL, the measurement setup from Figure 6 is used. The signal from the integrated bowtie antenna is picked up by a horn antenna (Mi-Wave 261G-10/387) with 12 dB gain. Thereafter, the signal is applied to a WR5 waveguide. As the spectrum analyzer (FSW from Rhode & Schwarz) has a span of 70 GHz, the RF signal is down-converted with the FS-Z220 Rhode & Schwarz mixer. The LO signal is generated from a signal generator (MG3690C from Anritsu) using a InP multiplier (SMZ170 from Rhode & Schwarz). All the measurement devices are available at the System-on-Chip Center (SoCC) at Khalifa University. For phase noise measurements, the spectrum analyzer has the phase noise option enabled by software. The measured spectrum of the PLL is presented in Figure 7. The resolution (and view) bandwidth is 47 KHz. The measured RF signal power is −19.53 dBm at 157.82 GHz. It can be observed that the worst-case scenario in-band spur for the standalone PLL is about −35.7 dBc. Therefore, the power at the PLL output after the buffer is close to 2 dBm, taking into account the gain of the PA, the gain of the two antennas, and the power losses due to free-space propagation at 160 GHz.

To measure the tuning range of the VCO, the PLL loop is disabled by turning off the components of the loop. The linear tuning range of the VCO is measured to be from 159.5 GHz to 167.5 GHz (as illustrated in Figure 8). Nevertheless, the VCO is functional at frequencies as low as 157.5 GHz. Additionally, it can be inferred by the figure that the measured VCO tuning frequency range closely converges with the simulated results.

Figure 9 shows the measured phase noise of the PLL, VCO, and reference. The measured phase noise of the PLL is −103 dBc/Hz @ 1 KHz and −128 dBc/Hz @ 100 KHz. The free-running VCO has a phase noise of −111 dBc/Hz @ 100 KHz and −132 dBc/Hz @ 1 MHz. The reference at 5 GHz has a phase noise of −110 dBc/Hz @ 1 KHz and −149 dBc/Hz @ 100 KHz.

Furthermore, (1) predicts a linear synchronization with the injection signal level of the injection-locked divider. This is confirmed by the measurements in Figure 10. Due to the limitation of the tuning range of the VCO, the last measured point (in blue) at −20 dBm input power is extrapolated from the previous measurement point.

Table 1 provides the benchmark of the proposed PLL and comparisons with other relevant works. For instance, [4] is a 198–274 GHz CMOS PLL. Even though the performance in terms of bandwidth, DC power consumption, and chip area of the PLL in this work is good, the phase noise lies above −80 dBc/Hz @ 100 KHz with an output power of only −11 dBm. Moreover, the proposed PLLs in [8] and [9] produce phase noises more than −90 dBc/Hz @ 100 KHz with DC power consumptions in the range of 323–380 and 1150–1250 mW, respectively. Meanwhile, the proposed PLL in this work has a phase noise of only −103 dBc/Hz @ 1 KHz and −128 dBc/Hz @ 100 KHz, which is among the lowest compared to the state-of-the-art sub-mm-wave PLLs. In addition, the bandwidth obtained is among the highest, with relatively high output saturated power in comparison to relevant works observed from Table 1. Although the performance of the designed PLL appears to be formidable, this was made possible by the use of the 22 nm FD-SOI CMOS, which offers relatively higher device performance at higher frequencies. As a result, the main disadvantage of the produced PLL is the high technology cost.

## 4. Conclusions

An 8 GHz linear tuning sub-mm-wave PLL applied in a transmitter front-end with antenna was designed and measured in this work. The proposed wideband sub-mm-wave I/Q VCO is based on a current tuning through differential voltage application, as it is more practical in the mm-wave frequencies and beyond. The PMOS-injected divider is based on a CPWG resonator and exhibits wider frequency locking range compared to the inductor-based NMOS frequency divider, covering the VCO bandwidth. The designed PLL was applied in a transmitter front-end along with an on-chip wideband bowtie antenna for performance testing purposes. The implemented PLL produces a phase noise of −128 dBc/Hz @ 100 KHz for the worst-case scenario, with outstanding performances in terms of power and area consumption compared to other recent relevant works (as represented by Table 1). To the knowledge of the authors, this is the lowest phase noise produced by a sub-mm-wave PLL to date.

## Figures and Tables

**Figure 1 micromachines-14-01010-f001:**
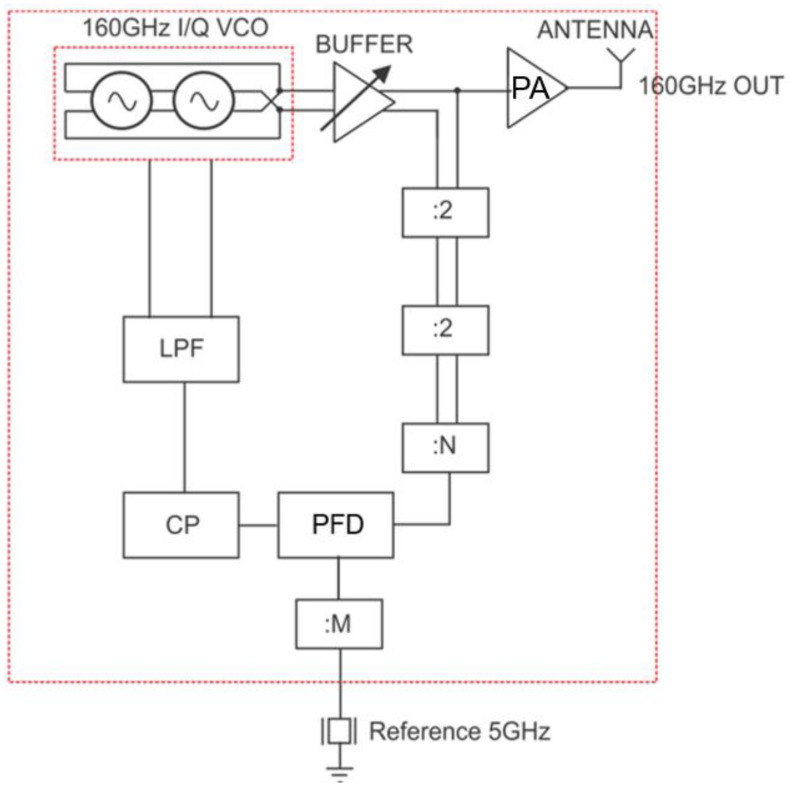
Block diagram of the proposed sub-mm-wave PLL.

**Figure 2 micromachines-14-01010-f002:**
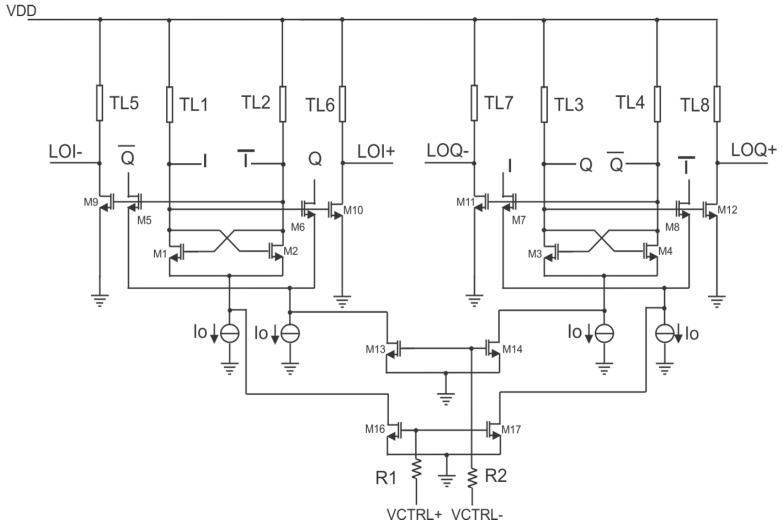
I/Q VCO circuit.

**Figure 3 micromachines-14-01010-f003:**
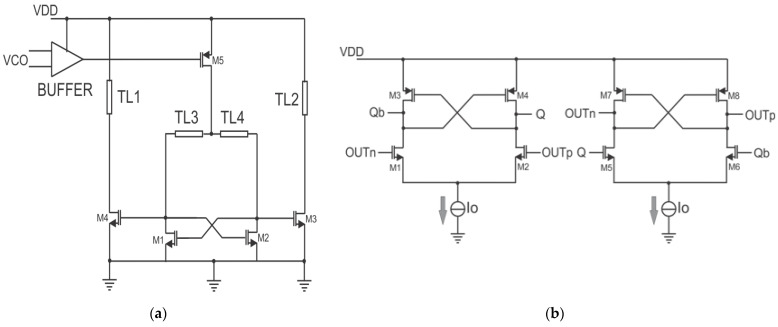
Frequency divider by 2: (**a**) injection-locked; (**b**) CML-based.

**Figure 4 micromachines-14-01010-f004:**
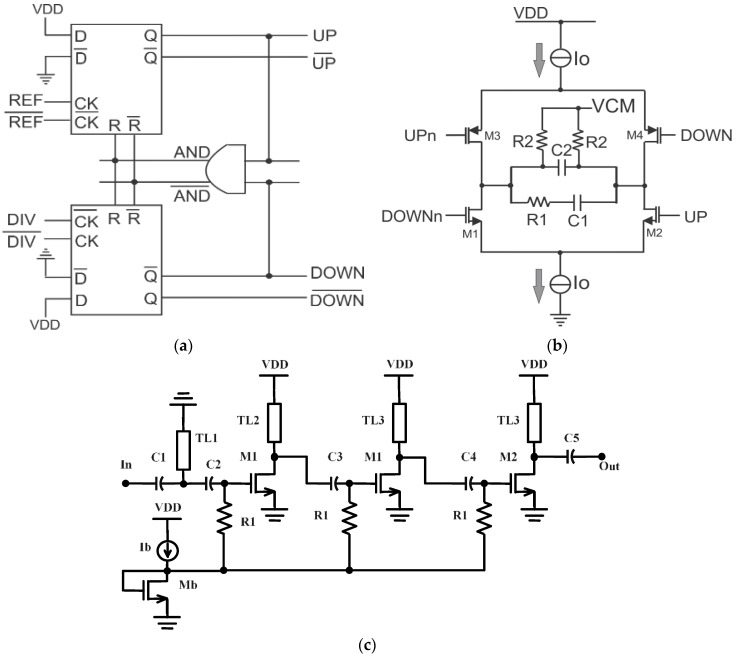
Schematic of: (**a**) PFD; (**b**) CP+LPF; and (**c**) PA.

**Figure 5 micromachines-14-01010-f005:**
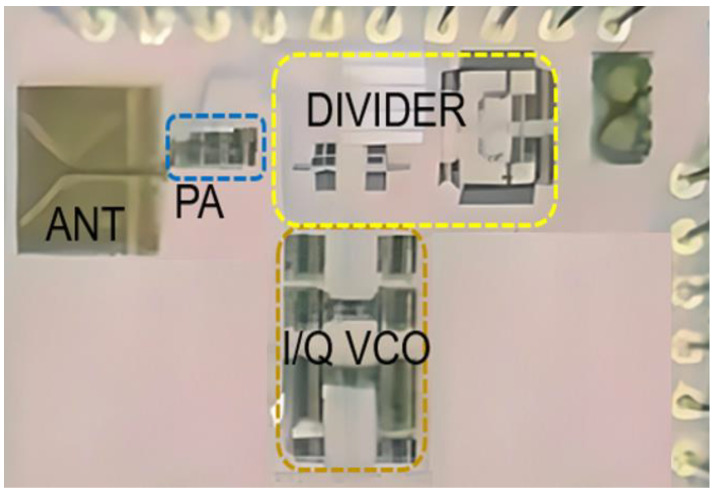
Chip photomicrograph (1.25 mm × 0.9 mm).

**Figure 6 micromachines-14-01010-f006:**
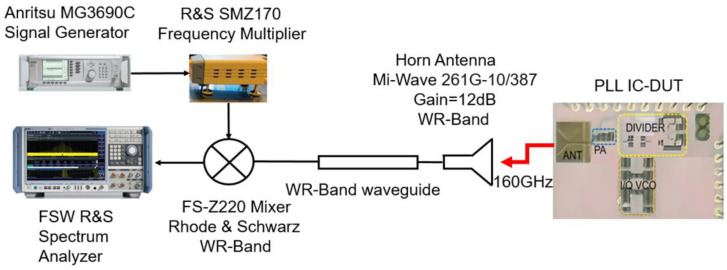
PLL measurement setup.

**Figure 7 micromachines-14-01010-f007:**
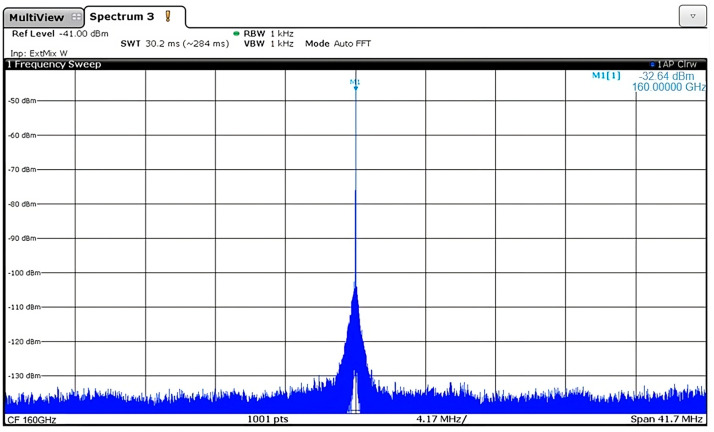
PLL output spectrum at 160 GHz.

**Figure 8 micromachines-14-01010-f008:**
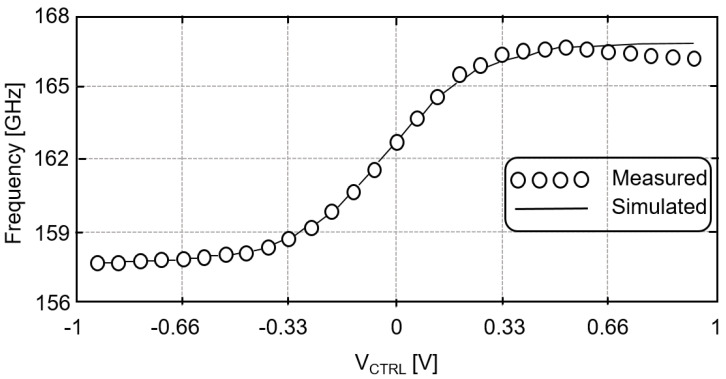
Measured versus simulated VCO tuning range.

**Figure 9 micromachines-14-01010-f009:**
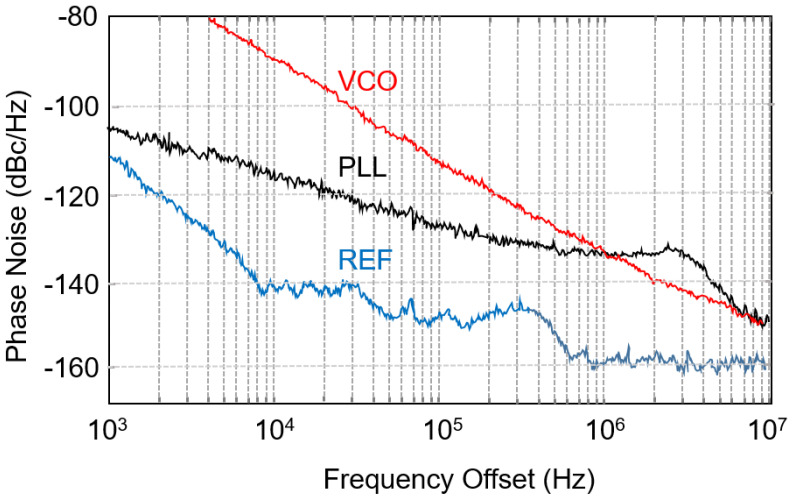
Measured PLL phase noise: closed loop (black), VCO (red), and reference (blue).

**Figure 10 micromachines-14-01010-f010:**
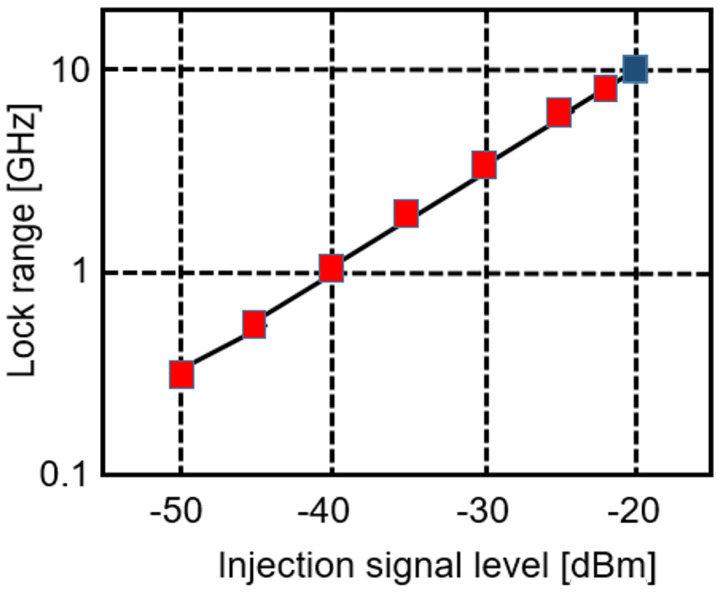
Measured synchronization range of the injection-locked divider @ 160 GHz.

**Table 1 micromachines-14-01010-t001:** Benchmark and comparison with other relevant works.

	This Work	[4]	[9]	[8]	[5]
Technology	22 nm CMOS	65 nm CMOS	0.13 µm SiGe BiCMOS	0.25 µm SiGe BiCMOS	0.13 µm CMOS
Frequency (GHz)	157.5–167.5	198–274	81–82 86–92 162–164	37.2–40	53–58
PN (dBc/Hz)	−128 @ 100 kHz	−78.2 @ 100 kHz	−89 @ 100 kHz, 79.4 GHz	−92.5 @ 100 kHz, 40 GHz	−85.2 @ 1 MHz
Output power (dBm)	2	−11	−25 @ 163 GHz	-	−37.85 @ 58 GHz
Power consumption (mW)	120.75	49.5	1150–1250	323–380	35.7
Chip area (mm^2^)	1.25 × 0.9	0.58	1.1 × 1.7	0.9 × 0.5	0.96 × 0.84

## Data Availability

The authors confirm that the data used in this study is either experimentally extracted and provided throughout the article or referenced below.

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
