# Peer review of "A Low-Phase-Noise 8 GHz Linear-Band Sub-Millimeter-Wave Phase-Locked Loop in 22 nm FD-SOI CMOS"

_micromachines, 2023, doi:10.3390/mi14051010_

Round 1

Reviewer 1 Report

This manuscript proposes A Low-Phase-Noise 8-GHz Linear-band Sub-Millimeter-Wave 2Phase-Locked Loop in 22-nm FD-SOI CMOS. The idea is good and experimentally verified; however, it may let down at its current form as it is very short and many informations should be addressed according to the following comments:

1- The abstract is very short, the contribution and the outcome of this work should be considered,

2- the introduction is very limited. Authors should introduce the topic and the research gap more extensively. The following references may be helpfull:

(1)doi: 10.3390/su141912066

(2) https://doi.org/10.1016/j.apacoust.2022.109129

(3) doi: 10.1109/LGRS.2022.3229556

(4) doi: 10.1117/1.AP.3.3.036003

(5) https://doi.org/10.1515/nanoph-2021-0801

3- Based on the comparision listed in table 1.  I did not see any improvment in the conducted work. Authors may need to explain the weaknes and the main advanatgious of proposed work.

4- Conclusion is not sufficent. It hshould be extended to cover th main idea and outcomes of this work.

5- References need to be updated and incresed. Majority are quite old.

Author Response

  • The abstract is very short, the contribution and the outcome of this work should be considered.

Response: The abstract was expanded by a couple sentences. This work mainly focuses on the performance of the implemented sub-millimeter-wave PLL in terms of noise and bandwidth, which appear to be outstanding compared to the state-state-of-art works.

  • the introduction is very limited. Authors should introduce the topic and the research gap more extensively. The following references may be helpfull:

(1)doi: 10.3390/su141912066

(2) https://doi.org/10.1016/j.apacoust.2022.109129

(3) doi: 10.1109/LGRS.2022.3229556

(4) doi: 10.1117/1.AP.3.3.036003

(5) https://doi.org/10.1515/nanoph-2021-0801

Response: The introduction was expanded to provide more details on the research gap. The highlighted research gap is the poor performance of sub-millimeter-wave VCOs in terms of noise and bandwidth

  • Based on the comparision listed in table 1. I did not see any improvment in the conducted work. Authors may need to explain the weaknes and the main advanatgious of proposed work.

Response: The main improvement generated by this work is the phase noise, which is the lowest compared to other relevant works. Moreover, the reported bandwidth and output power appear to be among the highest. A few sentences were added towards the end of section 3 (Experimental Results) stressing these advantages and the main disadvantage of this work, which is the high technology cost.

  • Conclusion is not sufficent. It hshould be extended to cover th main idea and outcomes of this work.

Response: A few sentences were added to the conclusion section.

  • References need to be updated and incresed. Majority are quite old.

Response: A few more references were added to the work.

Reviewer 2 Report

A few remarks on the manuscript:

1 - section 5 is indeed section 4.

2 - The last sentence on section 3 is repeated as the last sentence of the paper. It's redundant.

3 - Table 1 (Benchmark and comparison with other relevant works) is interesting and worth of exploring it. However the authors only write a paragraph, putting some emphasis on the citation of reference 4 values.  It would be welcome if some more comparisons could be written, namely the pros and cons of the proposed PLL, regarding the parameters on table 1.

Author Response

A few remarks on the manuscript:

1 - section 5 is indeed section 4.

Response: Indeed section 5 was meant to be section 4. The change is made in the manuscript.

2 - The last sentence on section 3 is repeated as the last sentence of the paper. It's redundant.

Response: Changes are made towards the end of section 3 to avoid this redundancy.

3 - Table 1 (Benchmark and comparison with other relevant works) is interesting and worth of exploring it. However the authors only write a paragraph, putting some emphasis on the citation of reference 4 values.  It would be welcome if some more comparisons could be written, namely the pros and cons of the proposed PLL, regarding the parameters on table 1.

Response: Changes are made in the last paragraph of section 3 to highlight the contribution of this work compared to other works in Table 1. Here, it is emphasized that the phase noise, bandwidth and output power obtained in this work are among the highest compared to other relevant works. The main disadvantage of the proposed PLL is the high cost due to the use of a deep sub-micron technology.

Reviewer 3 Report

The paper deals with a wideband linear differential tuning I/Q VCO with a very low phase noise base on Voltage-Controlled Oscillator (VCO), Current Mode Logic (CML) dividers, Injection-Locked Frequency Dividers (ILFDs),  a loop filter, and a Power Amplifier. The paper needs to be revised to fulfil the following advices:

·         Acronyms have to be introduced in the abstract section to help reader understanding the paper aim.

·         In Figure 7 you have to insert not only the worst case scenario output spectrum but also the most common one.

·         Authors have to insert Transient simulations and measurements of the output of VCO to rate the stability of the oscillator and of the design circuit.

·         Authors have to add the Magnitude of the transfer function of the PLL’s output noise with respect to different noise sources as a function of frequency.

·         Authors have to carry out a detailed noise analysis to support their results and justify them.

·         The main problem of this paper is the completely lack of designing process. No block has been dimensioned in terms of aspect ratio and capacitance for MOS, length for coplanar waveguides, gain for amplifier.

·         To stress the improvement with respect to the state of the art in terms of phase noise it could be useful to plot the phase noise vs frequency of different PLLs and discuss the results.

·         As regards the Introduction section, authors should discuss about other configurations of PLLs (see, i.e., A GHz spintronic-based RF oscillator. IEEE Journal of solid-state circuits45(1), 214-223.2009, Chip-scaled Ka-band photonic linearly chirped microwave waveform generator. Frontiers in Physics, 158, 2022) and expand the comparison area to other solutions. 

Author Response

The paper deals with a wideband linear differential tuning I/Q VCO with a very low phase noise base on Voltage-Controlled Oscillator (VCO), Current Mode Logic (CML) dividers, Injection-Locked Frequency Dividers (ILFDs),  a loop filter, and a Power Amplifier. The paper needs to be revised to fulfil the following advices:

  • Acronyms have to be introduced in the abstract section to help reader understanding the paper aim.

Response: Changes are made in the abstract accordingly. A few sentences were added to give it more details.

  • In Figure 7 you have to insert not only the worst case scenario output spectrum but also the most common one.

Response: This is a good point. However, authors no longer have access to the simulation tools and measurement devices used in this work as they were performed a few moths back and the main author has since left the institution where the work was performed. The worst-case scenario was considered as it helps in the comparison with other relevant works.

  • Authors have to insert Transient simulations and measurements of the output of VCO to rate the stability of the oscillator and of the design circuit.

Response: Authors no longer have access to the simulation tools and measurement devices used in this work.

  • Authors have to add the Magnitude of the transfer function of the PLL’s output noise with respect to different noise sources as a function of frequency.

Response: Authors no longer have access to the simulation tools and measurement devices used in this work.

  • Authors have to carry out a detailed noise analysis to support their results and justify them.

Response: The relevant detailed noise analysis was performed in the references [11] and [12], cited in the manuscript.

  • The main problem of this paper is the completely lack of designing process. No block has been dimensioned in terms of aspect ratio and capacitance for MOS, length for coplanar waveguides, gain for amplifier.

Response: In this work, the emphasis was made on the techniques used to improve the phase-noise and bandwidth of the PLL. Changes are made in section 2 to highlight this. For instance, the use of a differential tunning network to improve the performance of the VCO was discussed. Most of the blocks are conventional and therefore were not discussed in detail.

  • To stress the improvement with respect to the state of the art in terms of phase noise it could be useful to plot the phase noise vs frequency of different PLLs and discuss the results.

Response: Table1 provides the summarized performance measures of the relevant works. The reported data is either the best obtained by the respective authors or the most relevant to our work.

  • As regards the Introduction section, authors should discuss about other configurations of PLLs (see, i.e., A GHz spintronic-based RF oscillator. IEEE Journal of solid-state circuits, 45(1), 214-223.2009, Chip-scaled Ka-band photonic linearly chirped microwave waveform generator. Frontiers in Physics, 158, 2022) and expand the comparison area to other solutions.

Response: The introduction part was expanded to give more details on the research gap.

Reviewer 4 Report

1.      Most of the ideas written were already described in many literatures. The Authors tried to compile it but lack of the enhancement of the interrelation analysis between the references. It is advised that the authors give a deeper analysis on how these ideas become more applicative strategies so that they can contribute to the next step of implementation.

2.      Avoid lumping references. Instead summarize the main contribution of each referenced paper in a separate sentence and by including the reference number.

3.      The motivation of the paper is unclear, while it should be eye catching in order to make more sense. In this regard, a separate section on motivation and contribution should be included. The contribution of this work must be mentioned after the literature review section, not before.

4.      In the literature review, I believe the authors should provide a more critical review as they did not mention limitations of the mentioned papers.

5.      More in-depth analysis of the author's contribution of this paper in the introduction section.

6.      I would like to see more discussion of the literature so that I can clearly identify the article relates to competing ideas.

7.      The relevant discussion and content of the figures and tables in the text are not enough.

Author Response

  1. Most of the ideas written were already described in many literatures. The Authors tried to compile it but lack of the enhancement of the interrelation analysis between the references. It is advised that the authors give a deeper analysis on how these ideas become more applicative strategies so that they can contribute to the next step of implementation.

Response: The introduction section was expanded to highlight the research gap in the design of PLLs at sub-mm-wave frequencies. This work aims to address the issues of bandwidth and phase noise limitations in the design of sub-millimeter-wave PLLs. In this regard, the most common designs and techniques used in the literature were emphasized along with their limitations.

  1. Avoid lumping references. Instead summarize the main contribution of each referenced paper in a separate sentence and by including the reference number.

Response: Well noted. A couple reference lumping was done in the introduction part to stress a common method used in the literature.

  1. The motivation of the paper is unclear, while it should be eye catching in order to make more sense. In this regard, a separate section on motivation and contribution should be included. The contribution of this work must be mentioned after the literature review section, not before.

Response: The main motivation of this work is to address the research gap of sub-millimeter-wave PLLs, which are used for high data rate communication and imaging systems. The abstract section was modified to stress that. In addition, the main contribution of this work was summarized in the abstract, consisting of phase noise and bandwidth improvement of a PLL at sub-mm-wave frequencies.

  1. In the literature review, I believe the authors should provide a more critical review as they did not mention limitations of the mentioned papers.

Response: The introduction part was modified accordingly. The main limitations of the conventional PLLs are the limited bandwidth and high phase noise of the VCOs and the frequency dividers.

  1. More in-depth analysis of the author's contribution of this paper in the introduction section.

Response: Changes were made in the introduction part to highlight the research gap and the contribution of this work. Here, the issues related to the phase noise and bandwidth were addressed.

  1. I would like to see more discussion of the literature so that I can clearly identify the article relates to competing ideas.

Response: The introduction section was expanded to provide more details.

  1. The relevant discussion and content of the figures and tables in the text are not enough.

Response: Section 2, 3 and 4 were all expanded to provide more details. Moreover, the figure of the PA was added (Fig. 4 c)).

Round 2

Reviewer 3 Report

The Authors have modified the manuscript according to the Reviewer suggestions. However, it should be noted that, as reported in the first revision, Authors should discuss about other configurations of PLLs (see, i.e., A GHz spintronic-based RF oscillator. IEEE Journal of solid-state circuits45(1), 214-223.2009, Chip-scaled Ka-band photonic linearly chirped microwave waveform generator. Frontiers in Physics, 158, 2022and expand the comparison area to other solutions.  In the Authors' note, they report that this issue has been solved. However, no discussion and references are reported in the manuscript.

Author Response

Comments:

The Authors have modified the manuscript according to the Reviewer suggestions. However, it should be noted that, as reported in the first revision, Authors should discuss about other configurations of PLLs (see, i.e., A GHz spintronic-based RF oscillator. IEEE Journal of solid-state circuits45(1), 214-223.2009, Chip-scaled Ka-band photonic linearly chirped microwave waveform generator. Frontiers in Physics, 158, 2022) and expand the comparison area to other solutions.  In the Authors' note, they report that this issue has been solved. However, no discussion and references are reported in the manuscript.

Response:

The mentioned references have been added to the second paragraph of the introduction part (references [10] and [11]). [10] does not provide numerical values of crucial design parameters such the phase noise. Besides, the presented oscillator does not work at mm-wave frequencies. [11] is not CMOS integrated. Therefore, they were not added to the comparison table. The aim was to compare the performance of relevant standard CMOS PLL designs with our work.

Reviewer 4 Report

Authors have addressed my comments 

Author Response

Comments:

Authors have addressed my comments.

Response:

Thank you for your feedback.